# Serological and Molecular Evidence of Pathogenic *Leptospira* spp. in Stray Dogs and Cats of Sicily (South Italy), 2017–2021

**DOI:** 10.3390/microorganisms11020385

**Published:** 2023-02-02

**Authors:** Francesca Grippi, Vincenza Cannella, Giusi Macaluso, Valeria Blanda, Giovanni Emmolo, Francesco Santangelo, Domenico Vicari, Paola Galluzzo, Carmela Sciacca, Rosalia D’Agostino, Ilenia Giacchino, Cristina Bertasio, Mario D’Incau, Annalisa Guercio, Alessandra Torina

**Affiliations:** 1Istituto Zooprofilattico Sperimentale della Sicilia “A. Mirri”, 90129 Palermo, Italy; 2Rifugio Sanitario, Zona Industriale Prima Fase, Viale N. 7, 97100 Ragusa, Italy; 3Independent Researcher, Palermo, Italy; 4Istituto Zooprofilattico Sperimentale della Lombardia ed Emilia Romagna “Bruno Ubertini”, 25124 Brescia, Italy

**Keywords:** stray animals, leptospirosis, MAT, Real Time PCR, genotyping

## Abstract

Leptospirosis is a zoonosis of public health concern. Its prevalence in stray animals in the South of Italy is unknown. This study aimed to investigate *Leptospira* spp. prevalence in 1009 stray animals. Out of them, 749 were alive animals, including 358 dogs (316 from Palermo and 42 from Ragusa) and 391 cats (359 from Palermo and 32 from Ragusa), and 260 were corpses (216 dogs and 44 cats) randomly collected in Sicily. Dogs and cats underwent a serological screening by Microscopic Agglutination Test and a molecular investigation by Real-Time PCR targeting *lipL32*. Corpses were subjected to Real-Time PCR. Serological analyses showed a prevalence of 1.12% (4/358) for dogs and 0.26% (1/391) for cats, with the only positive cat coming from Palermo. Serogroup Icterohaemorrhagiae serovar Icterohaemorrhagiae or Copenhageni, followed by Canicola and Bratislava, were the most spread among dogs, while the serological positive cat reacted with Hardjo serogroup. Two urine (2/32, 6.25%) and one blood (1/391, 0.26%) samples of cats, all from Ragusa, were positive at Real-Time PCR for pathogenic *Leptospira* spp. Sequencing analyses showed the presence of *L. interrogans* serogroup Icterohaemorrhagiae serovar Icterohaemorrhagiae or Copenhageni in one of the positive urine samples and in the positive blood sample. Analyses on corpses showed a prevalence of 1.85% (4/216) in Sicilian dog kidney samples, while all corpses of cats resulted in negative. Genotyping analysis showed a genetic relatedness between cat and human isolates. Results show *Leptospira* spp. circulation among Sicilian stray animals. The genetic relatedness between cat and human isolates suggests a possible common infection source.

## 1. Introduction

Leptospirosis is an important zoonotic disease caused by infection with pathogenic spirocheaete bacteria belonging to *Leptospira* genus. Traditionally, the classification of Leptospires was based on their antigenic phenotype: more than 250 pathogenic *Leptospira* serovars are known. These are classified into 25 different serogroups. More recently, the biomolecular approach based on DNA homology has permitted us to identify 17 pathogenic species of *Leptospira* spp., seven of which (*L. interrogans*, *L. borgpetersenii*, *L. santarosai*, *L. noguchii*, *L. weilli*, *L. kirschneri*, *L. alexanderi*) are the most common agents causing disease in human and a wide variety of wild and domestic animals [1,2,3,4]. 

The infection is spread worldwide, particularly in Tropical areas, where many environmental and cultural conditions favorable for its transmission occur, such as the close coexistence between people and wild or domestic animals, as well as a predominantly warm-humid climate [5]. However, it can also occur in urban environments of industrialized countries or in temperate rural contests -with a seasonal trend characterized by a maximum incidence in warm and rainy months. Reservoir animals harbor Leptospires in their kidneys for a long time, often without symptoms, shedding them in the environment through urinary elimination and playing an important role as a source of infection [6,7]. Animals can develop variable clinical signs such as fever, renal and hepatic failure, and reproductive disorders [8]. As concerning the association of serovar-reservoir, many epidemiological studies reported specific animal species might act as the reservoir for particular *Leptospira* serovars [9].

Rodents are the main reservoir hosts of pathogenic *Leptospira* serovar Icterohaemorrhagiae, along with other marsupial and mammalian species [10]. Dogs are susceptible to a wide range of serovars. In Europe, the major ones detected in dogs are Icterohaemorrhagiae, Grippotyphosa, Australis, Sejroe, and Canicola [11]. Infection with one of these serovars frequently leads to severe clinical symptoms [12,13]. However, it is difficult to determine if susceptibility in dogs can be related to age, breed, or sex [14]. Among the main serovars detected in dogs, Canicola recognizes the dog as a reservoir host, representing a major concern in a One Health vision. On the other hand, mass vaccination of dogs has contributed to the decrease of infection by *Leptospira interrogans* serogroup Canicola serovar Canicola in Europe [12]. Different kinds of dog vaccines are available in Italy: bivalent vaccines containing two antigen strains, belonging respectively to two serovars (Canicola and Icterohaemorrhagiae), a trivalent vaccine containing three antigen strains, belonging to three serovars (Canicola, Icterohaemorrhagiae, and Grippotyphosa), and tetravalent vaccines containing four antigen strains, belonging to four serovars (Canicola, Icterohaemorrhagiae, Grippotyphosa, and Bratislava).

In cats, clinical signs are rare. The main serovars reported in Europe in these animals are Icterohaemorrhagiae, Canicola, Grippotyphosa, Pomona, Hardjo, Autumnalis, Ballum, and Bratislava. Probably cats become infected by catching reservoir animals, such as rats, mice, and voles infected with pathogenic strains of *Leptospira* [15]. However, no association has been found between serological positivity and sex or breed. Several studies reported instead an association with age, being a serological response to leptospira more frequent in old cats living outdoors or sharing the same household with another cat [16].

Moreover, both infected dogs and cats can shed Leptospires in the environment through their urine [16,17,18,19], potentially contributing to the infection spreading.

The Microscopic Agglutination Test (MAT) has been widely employed for the serodiagnosis of acute disease and is considered the gold standard to confirm leptospiral infection [20]. However, its partial inability to identify the infecting serovar and to differentiate the vaccine from natural exposure-induced titers have limited its use in a clinical setting [21]. PCR-based techniques have been successfully used to confirm infection at the early stages of the disease, as well as to identify infecting species by further sequencing of PCR amplicons. Currently, genotyping by multi-locus sequence typing (MLST) has been also used to establish *Leptospira* presence from various clinical samples [22,23,24]. However, PCR may not provide reliable results in the convalescent phase of the disease [25].

Leptospirosis still remains a major health problem and dogs and cats can play a sentinel role or can highlight the risk of exposure to humans. The aim of this study was to carry out a serological and molecular investigation to improve knowledge about the circulation of *Leptospira* strains in stray dogs and cats in Sicily.

## 2. Materials and Methods

### 2.1. Ethical Statement

All methods were carried out in accordance with relevant guidelines and regulations. The study did not involve any animal experiments. Sample collection was carried out for laboratory analyses and did not involve any suffering of the sampled animals. This study was conducted as part of the IZS SI 09/15 RC research project approved by the Italian Ministry of Health on 29 July 2016 (DGSAF-0018379-P-29/07/2016) and of the IZS SI 11/16 RC research project approved by the Italian Ministry of Health on 9 August 2017 (DGSAF-0018913-P-09/08/2017).

### 2.2. Study Design and Sample Collection

A total of 1009 biological samples from 574 stray dogs and 435 stray cats, were tested for this study between November 2017 and December 2021 (Table 1). 

Particularly, these two groups included: 358 asymptomatic alive dogs, out of which 316 samples were from Palermo city and province (North West Sicily) and 42 samples from Ragusa city and province (South Sicily); 391 asymptomatic alive cats, out of which 359 from Palermo area and 32 from Ragusa area. All the dogs from Ragusa were unvaccinated, like most of those from Palermo. Cats were all not vaccinated.

In addition, samples included 216 dog corpses and 44 cat corpses, randomly collected throughout the Sicilian territory and transferred to the Laboratory of Istituto Zooprofilattico Sperimentale (IZS) of Sicily for the diagnosis (Table 1). In particular, among dog corpses, one showed clinical signs attributable to leptospirosis. The signs included the typical jaundice color of the skin, and kidney and liver failure. 

Ethylenediaminetetraacetic acid (EDTA)-treated and untreated blood samples and urine samples (only from Ragusa) were obtained from each animal by public veterinarians during the sterilization, as part of the National Control Plan for the prevention of stray animals; while the kidney samples were collected from carcasses.

The blood samples were stored at 4 °C and centrifugated at 3000 rpm for 10 min at room temperature for serum separation. Serum samples were frozen at −20 °C until further serological analyses.

Samples from dogs and cats were subjected to both molecular and serological analyses, while only molecular investigations were made possible on corpses collected from the whole region. 

### 2.3. Serological Analysis

Serological investigation was carried out through MAT, using live antigens, kept as pure cultures in our Laboratory, for the detection of antibodies against the eight pathogenic serovars of *Leptospira* spp. circulating in Italy. Particularly, the following serovars were used as antigens: *L. interrogans* serogroup Australis, serovar Bratislava, strain Riccio 2; *L. borgpeterseni*, serogroup Ballum, serovar Ballum strain Mus 127; *L. interrogans*, serogroup Canicola, serovar Canicola, strain Alarik; *L. kirschneri*, serogroup Grippotyphosa, serovar Gryppotyphosa strain Duyster; *L. interrogans*, serogroup Icterohaemorrhagiae, serovar Copenhageni, strain Wjinberg; *L. interrogans*, serogroup Pomona, serovar Pomona, strain Pomona; *L. interrogans*, serogroup Sejroe, serovar Hardjo, strain Hardjoprajitno; and *L. borgpeterseni*, serogroup Tarassovi, serovar Tarassovi, strain Mitis Johnson.

Each antigen was a reference strain obtained by the National Centre for Leptospirosis IZS of Lombardia and Emilia Romagna “Bruno Ubertini” (IZSLER, Brescia, Italy) or by the Amsterdam University Medical Centers and was maintained in *Leptospira* Medium Base Ellinghausen-MacCullough-Johnson-Harris (EMJH—Difco, Becton, Dickinson and Company, Sparks, MD, USA), subcultured every 7–10 days and checked for purity, mobility and agglutination power before the use. The MAT was performed according to the World Organization for Animal Health (WOAH) guidelines [26,27]. Briefly, the MAT consists of a screening and a confirmation phase. In the first screening phase, all sera were diluted 1:100 using phosphate-buffered saline (PBS) at a pH of 7.6 and incubated for 2 to 4 h at 30 °C ± 2 °C with each *Leptospira* serovar in 1:1 proportion. The presence of microagglutination (positivity index) was verified by microscope observation in a dark field, with a 10× magnification, (Leica Microsystems, Wetzlar, Germany). Samples were considered positive when 50% or more Leptospires were agglutinated. MAT cut-off was 1:100. Therefore, each positive sample was successively titrated performing a series of double dilutions (from 1:100 to 1:6400) and incubating each dilution with the reactive serovar at 30 °C ± 2 °C for 2 h. The analysis was performed by microscopy observation. The titration was used to verify the last useful dilution for serological positivity. The serovar presenting the highest titer was considered the reactive serovar. Both in the screening and titration phase, four different controls were used: positive control (100% microagglutination) with positive sera for antibodies against the strains used; negative control (0% microagglutination) with negative serum for antibodies against *Leptospira* spp.; 50% microagglutination reading control for each antigen; antigen control (0% microagglutination) with only the antigens diluted 1:2 with physiological solution.

### 2.4. DNA Extraction and Real-Time PCR Detection

For DNA extraction from kidneys, the surface was flamed, and 1 g of tissue was withdrawn and homogenized in 9 mL of sterile physiological solution with Stomacher^®^ 80 Biomaster (Seward Limited, London, UK). For the urines, a volume of 2 mL was centrifuged at the rate of 12,000 *g* for 30 min and the pellet was resuspended with 0.2 mL of ultrapure distilled water.

DNA was extracted from 0.2 mL of blood or homogenized kidney or resuspended pellet from urines. Lysozyme 10 mg/mL (Roche, Basel, Switzerland) was added to each sample and incubated at 37 °C for 30 min as previously described [28]. DNA extraction was carried out using the PureLink Genomic DNA kit (Invitrogen, Paisley, UK), according to the manufacturer’s instructions. The extraction internal control (IC) included in the Quantifast Pathogen + IC Kit (Qiagen, Hilden, Germany) was used (0.1 µL per µL of elution volume). The DNA was stored at −80°C until use.

A multiplex Taqman-based Real-time PCR assay targeting the *Leptospira* genus-specific 16 S ribosomal RNA gene (rRNA gene) and the pathogen-specific *lipL32* gene present on the external membrane of pathogenic *Leptospira* species was performed, using primers described previously [29,30]. 

The PCR was performed in a 25 µL final volume, using 5 µL of extracted DNA, 5 µL of 5× Mastermix Quantifast (Quantifast Pathogen + IC Kit, Qiagen, Hilden, Germany), 700 nM of primers, and 200 nM of the probe. All extraction sessions included a negative control (water) and all amplification sessions included both a negative (water) and a positive control (DNA of *Leptospira interrogans* serogroup Australis serovar Bratislava strain supplied by the National Reference Center for Leptospirosis, IZS LER, Brescia, Italy and maintained at our laboratories). The assay was performed on a Bio-Rad CFX96 System using the following thermal conditions: a holding stage of 95 °C for 5 min, and 45 cycles of 95 °C for 15 s and 60 °C for 30 s.

Partial *rpoB* gene sequencing was performed for the identification of *Leptospira* species [31]. Analyses were conducted using the commercial GoTaq^®^ G2 DNA Polymerase (Promega Corporation, Milan, Italy) in a 25 µL reaction mix consisting of 5 µL of a 5× GoTaq^®^ Reaction Buffer, 1 µL of a dNTP mix (200 µM), 0.6 µL of each primer (0.5 µM), 0.125 µL of GoTaq^®^ G2 DNA Polymerase, and 5 of extracted DNA. Amplification was conducted under the following thermal conditions: 95 °C for 2 min to activate TaqPol followed by, 35 cycles of 94 °C for 30 s, 51 °C for 30 s, 72 °C for 30 s, and a final extension of 72 °C for 7 min and further sequenced (BMR Genomics, Padova, Italy) using the same amplification primer sets and analyzed using BioEdit Software version 4.0 [32].

The sequences of primers and probes are presented in Table 2.

Confidence intervals (CI_95%)_ of the positive results were calculated for proportions.

### 2.5. MLST Genotyping Analysis

Real-time PCR-positive samples were submitted to the IZSLER, Brescia for MLST analyses [33].

The MLST involves the sequence analysis of an internal region of about 400–600 bp of seven *housekeeping genes*, UDP-N-acetylglucosamine pyrophosphorylase (*glmU*), NAD(P)transhydrogenasesubunit alpha (*pntA*), 2-oxoglutarate dehydrogenase E1 component (*sucA*), triosephosphate isomerase (*tpiA*), 1-phosphofructokinase (*pfkB*), rod shape-determining protein rodA (*mreA*) and acyl-CoA transferase/carnitine dehydratase (*caiB*), as previously described by Boonsilp et al., 2013 [34]. The analysis was conducted using BioNumerics Software (version 7.6, Applied-Maths, Sint Maartens-Latem, Belgium), and sequence types (STs) were assigned through the Bacterial Isolate Genome Sequence Database (BIGSdb) (available online: https://pubmlst.org/leptospira/, accessed on 22 July 2022). The STs obtained, allow discrimination of the seven most common pathogenic species: *L. interrogans*, *L. kirschneri*, *L. borgpetersenii*, *L. noguchii*, *L. santarosai*, *L. weilii*, *L. alexanderi*.

A phylogenetic tree was built using MEGA X [35] on the concatamers of the seven MLST genes. The evolutionary history was inferred using the Neighbor-Joining method [36]. The evolutionary distances were computed using the Maximum Composite Likelihood method [37].

## 3. Results

### 3.1. Serological Results 

Serological analyses showed a prevalence of 1.12% (4/358, CI_95%_ 0.0003–0.022%), considering positive samples with titers higher than 1:100, while a prevalence of 0.25% (2/358, CI_95%_ −0.002–0.013%) is obtained considering only the positive dogs with titers higher than 1:400. In particular, all 42 dog serum samples collected from Ragusa area resulted negative at MAT for antibodies against pathogenic *Leptospira* strains included in the diagnostic panel. Instead, out of 316 dog sera samples from the Palermo area, 4 dogs (1.26% CI_95%_ 0.0003–0.024%) resulted positive at MAT against different serovars of *Leptospira interrogans* (cut-off ≥ 1:100). In detail, the following serogroups were detected: serogroup Icterohaemorrhagiae serovar Copenhageni with titers ranging from 1:100 to 1:1600; serogroup Australis serovar Bratislava, with titers ranging from 1:200 to 1:1600; serogroup Canicola serovar Canicola, with a titer of 1:3200. Antibody titers are detailed in Table 3. Serogroup Icterohaemorrhagiae serovar Icterohaemorrhagiae or Copenhageni, followed by Canicola and Bratislava, were the most spread among dogs.

Regarding cats, a prevalence of 0.26% (1/391, CI_95%_ −0.002–0.008%) was obtained. 

In particular, all 32 cats from the Ragusa area were negative at MAT for *Leptospira* pathogenic strains. One of 359 (0.28% CI_95%_ −0.002–0.008%) cats from the Palermo area resulted positive at MAT for *Leptospira interrogans* serogroup Sejroe serovar Hardjo, even if only with a titer 1:100 (Table 3). 

### 3.2. Molecular Detection and Genotyping Analyses 

All blood and urine samples collected from dogs both in Palermo and Ragusa areas resulted in negative by Real-Time PCR.

Regarding cats, two urine samples (2/32, 6.25% CI_95%_ −0.02–0.15%), from Cat B and Cat C, and one blood sample (1/391, 0.26% CI_95%_ −0.002–0.008%) from Cat D, all coming from Ragusa, was positive at Real-Time PCR for pathogenic *Leptospira* spp. (Table 4).

As concerning corpses, DNA of pathogenic *Leptospira* spp. was detected in 4/216 (CI_95%_ 0.005–0.036%) kidney samples collected from corpses of dogs (Table 4), including the one with clinical signs of leptospirosis (Dog H). The kidneys from all 44 cat corpses resulted in a negative.

### 3.3. Sequencing Analysis

For cats, sequencing analysis performed on one of the positive urine samples and on the positive blood showed the presence of *L. interrogans* serogroup Icterohaemorrhagiae. Sequencing analysis performed on the positive kidneys from the dog corpse showed the presence of *L. interrogans* serogroup Icterohaemorrhagiae. 

### 3.4. Genotyping Analysis

Genotyping analysis was carried out on the DNA of *Leptospira* extracted from the urine (Cat B, ID 3598/2019) and the blood (Cat D, ID 3606/2019) of two cats and from the kidneys of the dog (Dog H, ID 79788/2021) revealed the presence of strains belonging to ST17, referred to *L. interrogans* serogroup Icterohaemorrhagiae serovar Icterohaemorrhagiae or Copenhageni as showed in Figure 1.

Nucleotide sequences were submitted to the GenBank database under the following accession numbers: OP598830-OP598850.

Phylogenetic analysis carried out using the concatemer of the seven MLST genes showed that *Leptospira* strains circulate on Sicilian cats, clustered with the serogroup Icterohaemorrhagiae serovar Icterohaemorrhagiae (or Copenhageni).

Genotyping carried out on the positive dog kidneys revealed the identity of *L. interrogans* Icterohaemorrhagiae as shown in Figure 2, confirming MAT results.

The identified genotype ST17 indicated the presence of *L. interrogans* serogroup Icterohaemorrhagiae both in cats and dogs (Figure 3).

## 4. Discussion

This study investigated the seroprevalence of *Leptospira* antibodies and identified by molecular methods the infective *Leptospira* serovars in stray dogs and cats in southern Italy (Sicily).

Serological positivity emerged by using MAT in 4 of the 358 dogs in Sicily with titers ranging from 1:100 to 1:3200; serovars Copenhageni, Bratislava, and Canicola were prevalent, including the same strains of the tetravalent vaccine formulation in trade. The MAT is considered the serological test of choice for leptospirosis [40], despite its technical limitations, such as the inability to distinguish between present and past infection [41] and non-specific cross-reactions, such as those linked to a recent dog vaccination or recent infection [42]. A serological positivity with a minimum 1:100 antibody titer leads to a leptospirosis sospicious. 

In Italy, recent surveys conducted by MAT on dogs reported seroprevalence values ranging from less than 10% to over 20% [43,44]. In these studies, the most prevalent serogroups were Icterohaemorrhagiae, Grippotyphosa, Australis, and Canicola, and a low number of positive sera were also observed for Pomona and Sejroe serogroups.

*Leptospira* spp. presence on dogs was confirmed by Real-Time PCR: pathogenic strain DNA was detected in 4/216 kidneys from dog carcasses.

Sequencing analysis showed the presence of *L. interrogans* serogroup Icterohaemorrhagiae on the positive kidneys from the corpse sent with the clinical suspect of leptospirosis. Our results corroborate previous serological findings showing that Icterohaemorrhagiae is likely the main infecting serogroup in dogs [25,45], reinforcing the hypothesis that dogs are highly exposed to environmental contamination promoted by rodents. Serogroup Icterohaemorrhagiae is also a causative agent for human leptospirosis [46] and serovars Icterohaemorrhagiae and Copenhageni are reported as the main leptospiral causative agents in humans [47].

The vaccination against serogroup Icterohaemorrhagiae has been used for many years in Italy, contributing to preventing the onset of the disease in animals, however, as shown also by this study, the circulation of the pathogen is still high among dogs due to the role of rats as reservoirs. On the contrary, vaccination against *Leptospira interrogans* serogroup Canicola serovar Canicola allowed reducing notably the circulation of this serovar in Italy [48,49] as well as in the rest of Europe [12]. 

Regarding cats, one of 359 samples from Palermo resulted positive at MAT for *Leptospira interrogans* serogroup Hardjo serovar Sejroe, even if only with a 1:100 titer. In a previous study carried out in the Sicilian territory, Grippi et al., 2017 [50] found one cat sample (0.1%) positive for antibodies against *Leptospira* spp. with a MAT titer 1:100 for *Leptospira* Australis. Formerly, cat resistance against infections caused by spirochaetes was thought of without considering feline leptospirosis in the differential diagnosis in cats [13]. Antibodies detection in cats has shown infection can occur also among cats and these can be incidental hosts of some *Leptospira* serovars prevalent in wildlife or in other domestic animals [15], such as Ballum, Copenhageni, Hardjo, Icterohaemorragiae, Rachmati, Bratislava, Bataviae, Canicola, Autumnalis and Grippotyphosa [51].

Molecular investigations showed 2/32 cat urine and 1/391 cat whole blood samples positive for pathogenic *Leptospira* spp. DNA, all of them coming from the Ragusa area. Genotyping revealed that two of the positive cat samples were *Leptospira interrogans* serogroup Icterohaemorrhagiae serovar Icterohaemorragiae or Copenhageni. A similar study by Donato et al., 2022 [52] detected antibodies against serovars Poi, Bratislava, Arborea, Ballum, Pomona, and Lora in 15.3% (17/111) of cats (titers range: 20–320) and *Leptospira* spp. DNA in 3% (4/109) of blood and 9% (10/111) of urine samples.

The identified genotype ST17, indicating the presence of *L. interrogans* serogroup Icterohaemorrhagiae, was confirmed as the major cause of leptospirosis in cats and dogs. Previous serological surveys have consistently shown Icterohaemorrhagiae as the most reactive serogroup in dogs suspected of having leptospirosis [53]. 

The results also confirm the usefulness of using multiple diagnostic approaches to confirm leptospirosis. Indeed, the used Real-time PCR method can detect all pathogenic *Leptospira* species currently known [54,55] with efficiency, sensitivity, and specificity, allowing to overcome some limits of serological investigation (e.g., failure of an early diagnosis, low sensitivity). On the other hand, routine PCR does not allow for identifying the species/serovar involved, so serology has still a relevant role in epidemiological studies also allowing for cost containment. The importance of using multiple diagnostics is further strengthened by the occurrence of different phases during leptospirosis infection (leptospiraemia, leptospiruria, the last often intermittent), so PCR can lead to false negative results.

## 5. Conclusions

The results of this study show *Leptospira* spp. circulation among Sicilian stray animals. In particular, serogroup Icterohaemorrhagiae serovar Icterohaemorrhagiae and Copenhageni, followed by Canicola and Bratislava, were found in dog samples. One serological-positive cat reacted with the serogroup Hardjo, even if at a low titer. In addition, we detected two cat urine samples and a blood sample positive for pathogenic *Leptospira* spp. DNA. Sequencing analysis identified *L. interrogans* serogroup Icterohaemorrhagiae serovar Icterohaemorrhagiae or Copenhageni in one of the positive urine samples as well as in the blood sample. Genotyping analysis showed a genetic relatedness between cat and human isolates. Among corpses, *Leptospira* spp. DNA was detected in the kidneys of four dogs, including in the one with leptospirosis clinical signs. Genotyping carried out on this last sample revealed identity to *L. interrogans* serogroup Icterohaemorrhagiae serovar Icterohaemorrhagiae or Copenhageni.

To the best of our knowledge, this is the first report of *Leptospira interrogans* serogroup Icterohaemorrhagiae in cats in Italy. Further investigations will help understand the true role of *Leptospira* in cats. 

## Figures and Tables

**Figure 1 microorganisms-11-00385-f001:**
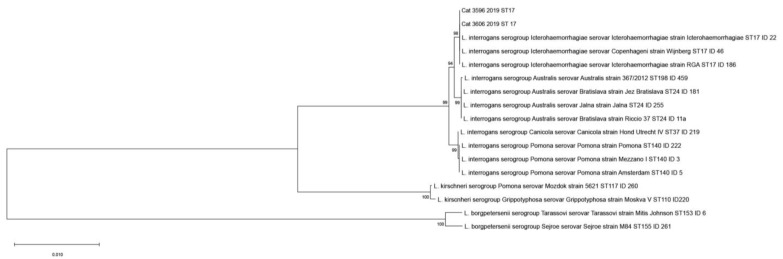
Phylogenetic tree based on concatenated sequences of the seven genes of the multi-locus sequence typing scheme (3111 bp) [34] for cats. Reference strains are indicated with: the sequence type number, the species, the serogroup, the serovar, and the name of the strain. The names of reference strains include the *Leptospira* species, serogroup, serovar, and strain.

**Figure 2 microorganisms-11-00385-f002:**
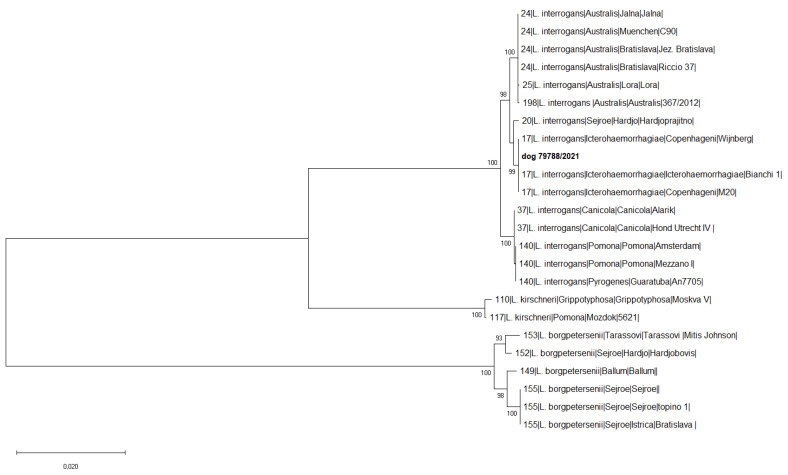
Phylogenetic relationship analysis based on concatenated sequences of the 7 MLST loci for dogs. Reference strains are indicated with: the sequence type number, the species, the serogroup, the serovar, and the name of the strain. The evolutionary history was inferred using the Neighbor-Joining method [38]. The percentage of replicate trees in which the associated taxa clustered together in the bootstrap test (1000 replicates) are shown next to the branches [38]. The tree is drawn to scale, with branch lengths in the same units as those of the evolutionary distances used to infer the phylogenetic tree [39]. There were a total of 3111 positions in the final dataset. Evolutionary analyses were conducted in MEGA X [35].

**Figure 3 microorganisms-11-00385-f003:**
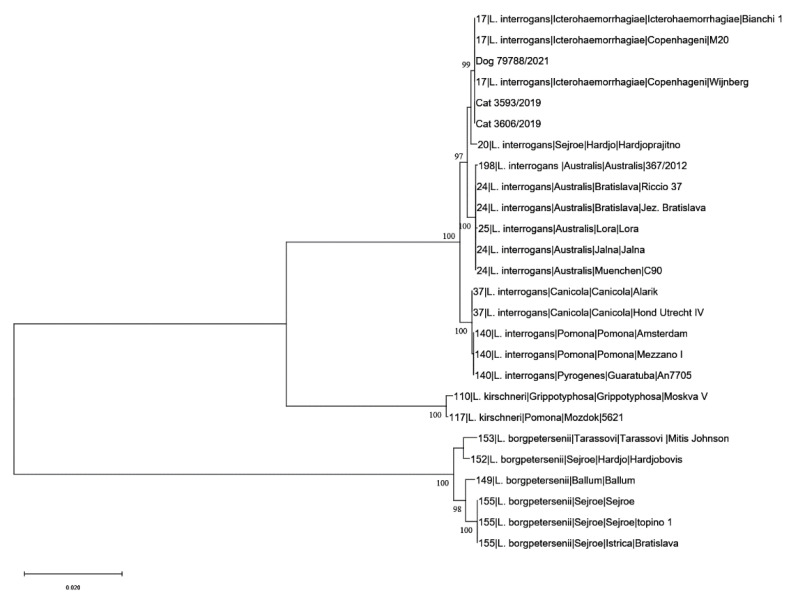
Phylogenetic relationship analysis based on concatenated sequences of the 7 MLST loci for dogs and cats. Reference strains are indicated with: the sequence type number, the species, the serogroup, the serovar, and the name of the strain.

**Table 1 microorganisms-11-00385-t001:** Samples collected and investigated for *Leptospira* spp. presence.

Sampled Species	Sample Type	Samples Number from Palermo Area	Number of Subjects Investigated from Palermo Area	Samples Number from Ragusa Area	Number of Subjects Investigated from Ragusa Area	Total Number of Subjects Investigated
Dog	Serum	316	316	42	42	358
	Whole Blood	316		42		
	Urine	Not available		42		
	Corpses	-		-		216
Cat	Serum	359	359	32	32	
	Whole Blood	359		32		391
	Urine	Not available		32		
	Corpses	-		-		44

**Table 2 microorganisms-11-00385-t002:** Molecular assays used in this study to detect and genotype *Leptospira* in cats and dogs.

Molecular Method	Primers Probes	Target	PCR Product Length	Reference
Real-Time PCR	LipL32-45F	*lipL32*	242 bp	[29,30]
	LipL32-286R			
	LipL32-189P			
	Lep-F	16S rRNA	173 bp	
	Lep-R			
	Lep-P			
Sequencing	Lepto 1900-F	*rpoB*	600 bp	[27]
	Lepto 2500-R			

**Table 3 microorganisms-11-00385-t003:** Antibody titers of serological positive dog and cat samples from Palermo Area.

Serogroup/Serovar	Dog A	Dog B	Dog C	Dog D	Cat A
Australis/Bratislava	1:800	1:1600	1:200	1:200	<1:100
Ballum/Ballum	<1:100	<1:100	<1:100	<1:100	<1:100
Canicola/Canicola	<1:100	1:3200	<1:100	<1:100	<1:100
Grypp./Grypp.	<1:100	<1:100	<1:100	<1:100	<1:100
Icteroh./Copenh.	1:100	1:1600	1:200	1:200	<1:100
Pomona/Pomona	<1:100	<1:100	<1:100	<1:100	<1:100
Sejroe/Hardjo	<1:100	<1:100	<1:100	<1:100	1:100
Tarassovi/Tarassovi	<1:100	<1:100	<1:100	<1:100	<1:100

**Table 4 microorganisms-11-00385-t004:** Results of Real Time-PCR, MLST, and Sequencing from positive samples obtained in this study. The sign “/” indicates not detected.

ID	Provenience	Type Samples	Real-Time PCR	Sequencing	MLST
Cat B	Ragusa	Urine	Positive	Positive	Positive
Cat C	Ragusa	Urine	Positive	/	/
Cat D	Ragusa	Blood	Positive	Positive	Positive
Dog E	Sicily	Kidney	Positive	Positive	/
Dog F	Sicily	Kidney	Positive	/	/
Dog G	Sicily	Kidney	Positive	/	/
Dog H	Sicily	Kidney	Positive	Positive	Positive

## Data Availability

The study did not involve any animal experiments. Blood, urine, and kidney samples were taken from animals to perform laboratory analysis without involving any suffering of the animals sampled.

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
