# Peer review of "Serological and Molecular Evidence of Pathogenic Leptospira spp. in Stray Dogs and Cats of Sicily (South Italy), 2017–2021"

_microorganisms, 2023, doi:10.3390/microorganisms11020385_

Round 1

Reviewer 1 Report

Dear Authors,

I read with great interest the paper, as leptospirosis is an important disease on both animal and human health sides. In dogs, the clinical illness can be critical and often deadly, while very little information are available on the infection in cats, traditionally considered resistant to the disease.

I have some comments to share, hoping to improve the scientific soundness of the paper.

I also recommend an ethical statement to accompany the paper.

Abstract:

In my opinion, the abstract could be improved to make it clearer. First, I suggest to start clarifying that dogs and cats from Palermo and Ragusa (specifying how many for each province) were subjected to a serological and molecular screening, while the corpses (how many) collected from all the Region were subjected only to a molecular screening.

Then, I would suggest better specifying that the result "cats for serogroup Hardjo" is referred to just one sample. About dogs, the results of the serology investigation highlight the Icterohaemorrhagiae serogroup, but also Bratislava and Canicola.

I strongly disagree with the statement (line 35-36) "genetic relatedness between cat and human isolates suggested that cats may be a maintenance host (...)". It seems more likely that cats and humans have been exposed to the same source of infection - the rat - well known as the reservoir host of serogroup Icterohamerrhagiae.

Moreover, Authors should clarify what they do mean with "reservoir" (line 38). Accordingly to the bibliography, reservoir hosts are animal species "adapted" to the pathogen (in case of leptospirosis, to a specific serovar), chronically infected, asymptomatic or paucisymptomatc, able to shed the pathogen for long periods of time. In my opinion, the results of this study do not show any of these characteristics, so I suggest drawing some more prudent conclusions.

In detail:

Line 22: the aim of the Authors was to investigate the "incidence" or the "prevalence"?

Introduction:

Line 58-59: “susceptible animals harbour L (…) for a long time”: I partially disagree; I would change “susceptible” to “reservoir”: Susceptible animals (and humans) can get ill and die, or recover, and frequently they do not harbour the pathogen after recovery if they are not adapted to the pathogen (reservoirs).

Line 60: Shedding, not shadding

Line 62-66: I suggest change to “bivalent vaccines containing two antigen strains, belonging respectively to two serovars” and so on…

Line 69: I suggest modifying to “its PARTIAL inability to identify the infecting serovar”. In the acute infection, many cross-reactions can occur due to IgM response, but after a few weeks/months, the antibody response become more specific and it can be useful to the diagnosis.

Line 86: “Animals can develop variable clinical signs depending on the infecting serovar”: this is probably true, but I can found very little references to support this topic, especially for dogs and cats. The symptoms here described are generally recognized for all the pathogenic leptospires, and probably represent only the “tip of the icerberg”.

Line 88-89: “A strictly association between serovar and animal species has been reported in many epidemiological studies”: I agree, but this is referred to the association serovar-reservoir. I found this statement not particularly linked to the previous one about symptoms. I suggest better specifying the mean of the sentence or the link between the two sentences.

Line 93: I suggest reminding that serovar Canicola recognizes dog as reservoir host, and can be more dangerous in a One Health vision. On the other hand, for the same reason and thanks to mass vaccination of dogs it can be controlled on an epidemiological point of view.

Line 98: Cats can become infected by catching rats, but also (and maybe more) mice, voles, etc.

Line 103-105: I agree infected dogs and cats can shed leptospires through urines, anyway carnivores’ urine are acid and rapidly inactivate the bacteria. There are very few data about the duration of the shedding in dog and cats, apart from serovar Canicola and dog. Moreover, there are some serological studies investigating the professional risk of veterinary clinicians, with no evidence of any professional risk. Therefore, I would suggest modifying the sentence “they play an important role in spreading of infection” in a more prudent statement.

Material & Methods

Line 117: please improve the description of sample collection, specifying how many samples came from Palermo and how many from Ragusa provinces. Furthermore, I suggest specifying that serological and molecular tests were performed on dogs and cats collected in Palermo and Ragusa, while only molecular investigations were made possible on corpses collected from the whole region.

Line 121-122: about corpses, in my opinion it is difficult to assess they were asymptomatic. Leptospirosis can cause hyperacute death and very aspecific clinical and anatomic-pathological syndromes. The different prevalence of infection between live animals (zero infected at molecular tests on blood+urine) and corpses (4/216 kidneys) does suggest something different.

Results

Line 219: the 42 dog sera collected in Ragusa were not described in M&M

Line 220: here we can see 316 sera collected in Palermo, but this number was not described in M&M.

Line 221: n. 4 or 5 dogs? In table 2 I can see 5 dogs.

Line 225: Canicola has no titre range, I see only one positive with titre of 1:3200

Line 234: how many blood and urine samples, in Palermo and in Ragusa?

Line 240: n. 2 urine samples and 1 blood samples from cats were positive, please report also the total number of examined samples. I cannot find the molecular results about the 44 cat’s corpses.

Line 247-249: L. interrogans should be in italics

Discussion

Line 302: please change australis to Australis

Line 305-308: I would suggest also considering these points: 1) routine PCR does not allow identifying the species/serovar involved, so serology can still help epidemiological studies at a lower cost than genotyping or microbiological cultures; 2) during infection, there are different phases: first leptospiraemia, then leptospiruria, and leptospiruria is often intermittent, so PCR can lead to false negative results.

Line 311: serological positivity emerged in 4 of 570 dog? Or 358?

Line 317-320: I would suggest explaining that Icterohaemorrhagiae is still circulating despite dog vaccination because of the rat acting as a reservoir. It is possible to compare the different epidemiological situation of Canicola, linked to dog as reservoir and more easily controlled by means of vaccination.

Line 325: please change kidney to kidneys

Line: 335-336: I would like to underline that the only one serological positive titre recorded in cat is of 1:100, not very significant.

Line 343: Authors report that 2/32 cat urine samples and 1/32 blood samples were positive. Is it the total of cats examined by PCR? I did not find this data specified in M&M. Are they the ones from Ragusa?

Conclusions

Line 357: again, I think it is difficult to assess that the death was not attributable to leptospirosis.

Line 361-364: if the dogs can be asymptomatic carriers, why no positivity was found on live animals? Why the prevalence is different between live and dead dogs? Please comment this point.

Line 364-365: with regards to the antibody response to Icterohaemorrhagiae: again, humans and dogs can be exposed to the same source of infection (rats). We cannot conclude that dogs can infect humans.

Line 376-377: again, I would be suggest to be prudent about the capability of cats to be a source of contagion. Acid urines, the amount of shedding, and its duration should be considered too.

Reviewer 2 Report

Revision of manuscript microorganisms-2138397

Dear Authors,

Your manuscript entitled “Serological and molecular evidence of pathogenic Leptospira spp. in stray dogs and cats of Sicily (South Italy), 2017-2021” describes a serological and molecular survey on Leptospira in dogs and cats.

The obtained data are interesting, but the manuscript needs a restyling.

In particular:

·         Discussions and conclusions must focus more on the obtained results.

·         Your data are valuable, but you find few positive animals and it did not emerge alarming situation (considering tested animals aren’t pets), be careful with strong statements.

·         English should be revisited. In my opinion, but I am not a native specker, it is not a problem of grammar or language, but also of stile and presentation; some parts are hard to read and follow; this does not valorize your interesting results.

Below You can find some more detailed comments and suggestions:

·         Abstract

o   should be reformulated

o   “…cats may be a maintenance host for serogroup Icterohaemorrhagiae and a source for human infection” NO.

·         Introduction

o   Lines 49-50: these are species and not strains; furthermore the number of pathogenic species, belonging to subclade P1, is 17 (Vincent, A.T.; Schiettekatte, O.; Goarant, C.; Neela, V.K.; Bernet, E.; Thibeaux, R.; Ismail, N.; Khalid, M.K.N.M.; Amran, F.; Masuzawa, T.; et al. Revisiting the taxonomy and evolution of pathogenicity of the genus Leptospira through the prism of genomics. PLoS Negl. Trop. Dis. 2019, 13.).

o   Lines 58-61: here you are specking about maintenance hosts only, not about all the susceptible host.

o   Lines 62-66: are Authors speaking about dog vaccines only? Please specify.

o   Lines 82-85: is this relevant?

o   Lines 82-105: Why did Authors jump from host to diagnostic techniques and back to hosts? Please rearrange introduction organization.

o   Line 90: Leptospira serovar (or serogroup) Icterohaemorrhagiae.

o   Lines 92-93: serovars not in italics.

o   Lines 97-98: serovars not in italics.

o   The aim is incomplete, Authors did not perform only sero-epidemiological analyses, but molecular investigations too; please correct.

·         Material and methods

o   Line 125: vaccinated.

o   Lines 129-130: this information is not necessary here; you described DNA extraction below.

o   Lines 137: serovars instead strains.

o   Lines 137-143: add strains employed.

o   Lines 156-157: ok, but this information should be provided below, after describing titer evaluation.

o   Lines 168-170: add information about urine samples processing for DNA extraction.

o   Lines 172-175: LipL32 without capitol letter (lipL32). Furthermore, as you reported, 16 S ribosomal RNA gene detected all leptospire (Leptospira genus) and lipL32 detect only pathogenic strains, so the sentence was performed to detect intermediate and pathogenic leptospires is wrong, because you could find saprophytic strains too. Please correct.

o   Line 204: sucA in italics.

·         Results

o   Line 222: … at MAT against different serovars of Leptospira interrogans.

o   Lines 229-230: serogroup Sejroe, serovar Hardjo.

o   Line 247 and 248: L. interrogans in italics.

o    

·         Discussion

o   Line 308: LipL32 without capitol letter (lipL32).

o   Lines 311-312: titers range were repeated 2 time in these 2 rows.

o   Lines 317-319: this sentence has not a lot of sense; vaccination prevent the diseases and eventually the infection by the included serovars, but it can not help to reduce the circulations of some serovars like Icterohaemorrhagiae that is carried by small rodents.

o   Lines 329-331: rewrite this sentence.

o   Line 245: serovar instead serovariant.

o   Considering that from line 286 to lines 310 Authors performed only general comments, some more discussion on obtained results are expected.

·         Conclusions

o   Lines 362-364: with obtained results you can not reach this strong conclusion.

o   Lines 364-366: this is not a conclusion of your study.

o   Lines 368: this information can not be reported for the first time in conclusion section.

o   Lines 369-371: rewrite this sentence.

o   “(typical leptospiruria of the "reservoir host").” ?

o   Lines 377-380: Authors found only 1 cat positive by PCR, you can not state this.

o   Lines 383-384: rewrite this sentence .

o   All the conclusion section should be rearranged, focusing on obtained results, and reformulated.

I sincerely hope that these suggestions will enhance this manuscript. However, if I have made any errors or misinterpretations, I apologize in advance.

Sincerely

The Reviewer

Round 2

Reviewer 1 Report

Dear Authors,

Thank you for answering and welcoming the comments from the two reviewers- some of them were fortunately in agreement with each other.

I have still some minor comments; I hope you will appreciate them in order to further improving the quality of the paper. I am sorry, but I found some further inaccuracies.

About the ethical statement, I suggest adding an official opinion of an Ethical Committee about the study.

Line 42-42: “The two cats excreted L. interrogans serogroup Icterohaemorrhagiae serovar Icterohaemorrhagiae or Copenhageni in the urine.” This is not coherent to the results section, where serovar Icterohaemorrhagiae or Copenhageni were identified in one urine sample and one EDTA blood sample. Two urine samples resulted positive to PCR, but only one were characterized, is it right?

Line 130-131: “Several studies reported instead an association with age, being leptospirosis more frequent in old cats living in outdoor (…)” I suggest to change to “Several studies reported instead an association with age, being serological response to leptospira more frequent in old cats living in outdoor (…)”

Line 316: why “positive samples with titers higher than 1:200?” when the cut-off is 1:100, as stated at line 236.

Line 320: “all 42 dog serum samples collected from Ragusa area, resulted 318 negative at MAT for antibodies against pathogenics Leptospira strains”: I suggest changing to “all 42 dog serum samples collected from Ragusa area, resulted 318 negative at MAT for antibodies against pathogenics Leptospira strains included in the diagnostic panel”

Line 420: “the inability to distinguish between present and past infection and non-specific cross reactions, such as those linked to a recent dog vaccination; I suggest changing to “the inability to distinguish between present and past infection and non-specific cross reactions, such as those linked to a recent dog vaccination or recent infection;

Line 422: “antibody 421 titers >1:400 confirm a diagnosis of canine leptospirosis.” I do not agree to this statement, as we cannot distinguish between present and past infection (line 420). We need to demonstrate a three to fourfold titer increase, or we can only suspect canine leptospirosis.

Line 542: again, “we detected two cats excreting in the urine L. interrogans serogroup Icterohaemorrhagiae serovar Icterohaemorrhagiae or Copenhageni” is not in agreement with the results description.

Line 546: “Among corpses, Leptospira spp. DNA was detected in the kidneys of dogs” I suggest indicating the number.

Author Response

Please see the attachment regarding the point-by-point response to the reviewer’s comments. 

Reviewer 2 Report

No more comments. Good work!